# Show Me Your Friends and I Tell You Who You Are: The Many Facets of Prion Protein in Stroke

**DOI:** 10.3390/cells9071609

**Published:** 2020-07-02

**Authors:** Berta Puig, Denise Yang, Santra Brenna, Hermann Clemens Altmeppen, Tim Magnus

**Affiliations:** 1Neurology Department, Experimental Research in Stroke and Inflammation (ERSI), University Medical Center Hamburg-Eppendorf, 20246 Hamburg, Germany; denise.yang@stud.uke.uni-hamburg.de (D.Y.); s.brenna@uke.de (S.B.); t.magnus@uke.de (T.M.); 2Institute of Neuropathology, University Medical Center Hamburg-Eppendorf, 20246 Hamburg, Germany; h.altmeppen@uke.de

**Keywords:** prion protein, stroke, ischemia, neuroprotection, regeneration, extracellular vesicles

## Abstract

Ischemic stroke belongs to the leading causes of mortality and disability worldwide. Although treatments for the acute phase of stroke are available, not all patients are eligible. There is a need to search for therapeutic options to promote neurological recovery after stroke. The cellular prion protein (PrPC) has been consistently linked to a neuroprotective role after ischemic damage: it is upregulated in the penumbra area following stroke in humans, and animal models of stroke have shown that lack of PrPC aggravates the ischemic damage and lessens the functional outcome. Mechanistically, these effects can be linked to numerous functions attributed to PrPC: (1) as a signaling partner of the PI3K/Akt and MAPK pathways, (2) as a regulator of glutamate receptors, and (3) promoting stem cell homing mechanisms, leading to angio- and neurogenesis. PrPC can be cleaved at different sites and the proteolytic fragments can account for the manifold functions. Moreover, PrPC is present on extracellular vesicles (EVs), released membrane particles originating from all types of cells that have drawn attention as potential therapeutic tools in stroke and many other diseases. Thus, identification of the many mechanisms underlying PrPC-induced neuroprotection will not only provide further understanding of the physiological functions of PrPC but also new ideas for possible treatment options after ischemic stroke.

## 1. Introduction

According to the World Health Organization (WHO), stroke is the second cause of death in the world—with almost six million deaths—and the third global cause of disability (https://www.who.int/gho/mortality_burden_disease/causes_death/top_10/en/) [1]. About 87% of all strokes are ischemic, while 10% are caused by intracerebral and 3% by subarachnoidal hemorrhages [1]. In ischemic stroke, a vessel is occluded, leading to loss of blood flow and, consequently, to a lack of glucose and oxygen supply in the affected brain area. This triggers complex pathophysiological mechanisms involving several events that can last for days. Hence, neurons in the ischemic core region are irreversibly damaged and undergo oncotic/necrotic cell death [2]. Surrounding the ischemic core, there is an area termed penumbra that still receives limited blood flow from the neighboring non-affected areas. Here, neurons are metabolically active but electrically silent and can be rescued if the blood flow is restored soon after the insult. However, neurons in the penumbra are exposed to excitotoxicity (due to the massive release and accumulation of glutamate into the extracellular space and the consequent Ca2+ entry into the cell), oxidative stress (due to the production of nitrogen and oxygen reactive species (NRS and ROS)), spreading depolarizations, and inflammation, factors that will determine death or survival of these neurons [3,4,5,6]. Rescuing and minimizing the penumbra area is one of the goals in stroke therapy because it will improve the clinical outcome [7,8]. Currently, the two treatment options for patients with acute ischemic stroke are focused on reperfusion either by intravenous thrombolysis or by endovascular thrombectomy. The former can only be administered within 4.5 h after symptom onset due to the secondary effects [9], while the latter has an extended time window of 24 h from symptoms’ onset, depending on clinical and imaging criteria [10,11]. When used in combination, it further improves functional outcome [12]. However, as not all patients are eligible for one of these treatments, there is an urgent need for new treatment options.

On the other hand, several endogenous self-recovery mechanisms take place after stroke, such as angiogenesis, remodeling of the extracellular matrix (ECM), synaptogenesis, neurogenesis, and gliogenesis, whose stimulation may be a therapeutic target [13,14,15].

The cellular prion protein (PrPC) is a cell surface glycosylphosphatidylinositol (GPI)-anchored glycoprotein encoded by the *PRNP* gene, which is highly expressed in neurons but also other cells and tissues [16]. Its structure comprises a disordered *N*-terminal part (flexible tail) featuring an octapeptide repeat region, a charged and a hydrophobic domain, and a *C*-terminal globular domain that includes two *N*-glycosylation sites and a GPI anchor for attachment to the outer leaflet of the plasma membrane [17,18]. In its misfolded form (PrPSc), PrP is the main component of the prion particle, the causative and transmissible agent of prion diseases. These are progressive fatal neurodegenerative disorders affecting both humans and animals [19]. The causes of PrP misfolding include sporadic conformational change, genetic mutations, or contact with infectious material already containing misfolded seeds leading to PrPSc-templated conversion of PrPC [20]. PrPC is very well conserved in mammals [21,22], and some domains (i.e., the C-terminal domain) are even preserved from fishes to mammals, probably indicating an important physiological function [23,24,25,26]. However, a clearly defined function is not yet specified but several, such as a role in regulating synaptic activity, as a receptor for toxic oligomers, as a regulator of neuritogenesis, and in contributing to peripheral myelination (reviewed in [27]. An explanation for this myriad of functions is probably related to both PrP’s various binding partners) [28] and its proteolytic processing leading to several biologically active fragments [29]. This diversity (and complexity) is further increased by the fact that PrPC is released by cells of various tissues in association with extracellular vesicles (EVs).

In the last two decades, several studies have pointed out a protective role of PrPC in ischemic damage. In this review, we summarized the role of PrPC in different pathophysiological aspects of stroke and discussed therapeutic possibilities. Although PrPC has been implicated in many processes, some as important as inflammation [30] and blood–brain barrier (BBB) integrity [31,32], we here focused on its role in signaling, stem cell homing, and EV delivery upon ischemic damage.

## 2. Evidence That PrPC Has a Protective Role after Ischemic Damage

First studies with PrP knock-out mice (PrP0/0) did not shed much light on possible function(s) of PrPC because of their normal development and lack of any gross abnormal phenotype in development and behavior [33]. However, it was suggested early on that PrP0/0 cells were more vulnerable to oxidative stress. Several reports advocated for an antioxidant role of PrPC by regulating both the Cu/Zn superoxide dismutase ((SOD) given PrP‘s ability to bind Cu2+ with high affinity [34,35,36,37,38] and glutathione reductase activity [39], although the later could not be confirmed in vivo [40]. Additional studies also pointed to a protective role against oxidative stress. Prion-infected mice showed increased levels of oxidative stress, most likely as a consequence of PrPC loss of function [41]. Fittingly, PrP0/0 mice had reduced protection against ROS [42,43]. Based on these results, McLennan et al. hypothesized that PrPC expression could be upregulated in diseases where oxidative stress played an important role. They performed immunohistochemistry in post-mortem human brain tissue from adults with cerebral ischemia (CI) and perinatal hypoxic-ischemic injury (HII). They found, indeed, an increased PrPC reactivity in the penumbra and hypoxic regions in both instances, together with elevated PrP mRNA levels during hypoxia in brain tissue of HII. The latter reinforced the idea of de novo synthesis of PrPC instead of just a simple malfunction in protein turnover or cellular transport. The authors also showed upregulation of PrPC in the penumbra area in a mouse model of permanent cerebral ischemia ((pMCAO) middle cerebral artery occlusion). PrP0/0 mice subjected to pMCAO had a larger ischemic area than wild-type (WT) mice, an effect that was directly dependent on the amount of PrPC, as confirmed by rescue of the phenotype in heterozygous mice (PrP+/0) [44]. Mitsios et al. also studied human brain samples and found similar results, with increased levels of PrPC in the peri-infarcted area in neurons and endothelial and inflammatory cells, coinciding with neuronal survival. Increased levels of PrPC circulating in plasma were also found 24 h after stroke onset. The pMCAO in rats caused increased PrPC levels in the penumbra, and oxygen-glucose deprivation (OGD) in human fetal neurons (HFN) likewise induced an increase in PrPC [45]. In support of this, elevated PrPC levels were also found in another model of stroke in rats (photothrombotic model, with permanent occlusion of small vessels) 4 h after stroke in the penumbra using proteomic analysis [46]. Weise and colleagues, by comparing the pMCAO model with the milder transient ischemic damage ((tMCAO) with vessel occlusion of 60 min), observed that in the severe model PrPC was upregulated in the hypoxic area between 4 h and 8 h after the insult, returning to normal levels at 24 h, whereas the transient model did not induce a significant upregulation of PrPC levels [47]. Along this line, Shyu et al., using a rat model of tMCAO ligation (for 90 min) and reperfusion, observed an increase of PrPC expression in neurons, glia, and endothelial cells in the penumbra which was linked to neuroprotection [48]. Taken together, these results indicate that PrPC upregulation could be a survival mechanism in the penumbra triggered early after severe damage, which is no longer needed once the tissue gets reperfused.

Other studies using PrP0/0 mice subjected to either p- or tMCAO consistently revealed greater brain damage compared to WT [44,49,50,51]. Moreover, mice lacking Shadoo (a protein belonging to the mammalian prion protein family that structurally resembles the *N*-terminal half of PrPC) at least showed a tendency to also be more susceptible than WT 24 h after tMCAO [52]. The obvious question arose: Is PrPC overexpression protective? In a cell model of hypoxia-reoxygenation, overexpression of PrPC led to increased neuroprotection [48]. In the same study, intracerebral injection of a plasmid coding for PrPC improved behavior and lessened the ischemic damage in rats subjected to 90 min of MCAO and bilateral CCA (common carotid arteries) ligation. While beneficial effects by decreasing the volume of ischemic damage were confirmed in human PrPC-overexpressing (TG35) compared to WT mice [53], a contrasting report did not find any effect resulting from mouse PrPC overexpression (*tga20* mice) [49]. One plausible explanation for these seemingly controversial findings (apart from the use of different PrPC species transgenes) is that the duration of tMCAO was different in the two latter studies (90 min vs. 30 min), possibly accounting for a different degree of damage. This could support what is indicated above, namely that increased PrPC is primarily required soon after severe damage. It can also be that amounts of PrPC modulate the long-term recovery, given that *tga20* mice, compared to WT and PrP0/0 mice, showed neurological improvement and decreased stroke volume after 28 days of reperfusion following tMCAO (45 min) [54].

Finally, PrPC seems to be protective in hypoxic conditions not only in the brain but also in other organs. Hearts from WT mice with permanent coronary artery ligation show a transient increase of PrPC. In a model of ischemia/reperfusion, overexpression of PrPC was highly protective compared to WT, where basal amounts of PrPC could not counteract the aggressiveness of the treatment and presented comparable levels of cell death as PrP0/0 mice. In that study, protection conferred by PrPC overexpression resulted in reduced ROS-related cell damage [55]. A similar situation was observed in the kidney, where PrP0/0 mice showed more tubular damage and increased cell death related to ROS than WT [56]. The reported upregulation of PrPC in gastric cancer cell lines may also protect them from hypoxia [57].

As judged from the experiments described above, PrPC clearly plays an important role in (neuro)protection, as observed in mouse and rat models of ischemia and in humans after stroke. The next sections will address in more detail the different mechanisms whereby PrPC may exert its protective role(s).

## 3. An Overview of PrP-Associated Signaling Pathways and Their (Potential) Role in Ischemia

The plasma membrane is not evenly distributed and there are subdomains known as lipid rafts or detergent-resistant domains ((DRMs) because of their property of being insoluble in mild detergents [58]) enriched in saturated sphingolipids and cholesterol. Because of their property of selectively including and excluding certain proteins, these domains can recruit adaptors, enzymes, and scaffolding proteins after ligand binding to receptors, thus constituting important signaling platforms [59]. PrPC is mainly present at the surface of the plasma membrane and its GPI anchor drives localization within lipid rafts. By this fact, a growing body of evidence indicates that PrPC acts as a receptor or co-receptor and mediates cellular signaling involving different signaling pathways [17,60,61,62,63,64]. It has been suggested that PrPC is a scaffolding protein that belongs to a multiprotein complex able to modulate several functions, probably in a cell- and context-dependent fashion [65,66]. However, it is possible that some of these many functions are just a consequence of the use of PrP knock-out mouse models containing uncontrolled genetic pitfalls [27]. In this review, we will, therefore, highlight when a suggested function was not only found in mice but also in other animal models, cells, and/or in human brains, as this would increase the probability of being a genuine role for PrPC.

Several lines of research have focused on the study of signaling cascades activated by PrPC. Importantly, because PrPC does not contact the cytosol, it must bind certain partners that structurally traverse the plasma membrane and have a cytosolic tail to exert a signaling function (reviewed by [67]). In the next paragraphs, we will dissect some of these PrPC-related signaling pathways that could mediate its role upon ischemic damage.

### 3.1. The Mitogen-Activated Protein Kinase (MAPK) Pathway

The MAPKs are serine/threonine protein kinases that mediate complex signal transduction from a wide variety of cellular processes ranging from proliferation, differentiation, migration, cell death/survival, and environmental stress response. They comprise four main subfamilies: The extracellular signal-regulated kinases (ERK1/2), the c-Jun NH2-terminal kinases (JNK 1, 2, and 3), the p38 (α, β, γ, and δ) kinases, and the ERK5 (also known as big MAPK-1, BMK1) subfamilies [68,69,70].

MAPKs were early on implicated in signaling after ischemic brain damage and consistently found upregulated in cells of the penumbra in tMCAO, pMCAO, and global ischemia animal models, with different participation in cell death or survival depending on the cell type and insult [71,72,73,74].

PrPC elicits signaling through MAPKs. The first indications came from in vitro experiments. Thus, antibody-mediated clustering of PrPC in neuronal cells lines 1C11 (that can be differentiated to serotoninergic or noradrenergic neurons) and GT7 (hypothalamic cells) or in a lymphocyte cell line promoted nicotinamide adenine dinucleotide phosphate (NADPH) oxidase-mediated ROS increase, which led to a transient increase in ERK1/2 phosphorylation. Interestingly, only in differentiated 1C11 cells, Fyn kinase (see below) was upstream of either NAPDH oxidase activation or ERK1/2 phosphorylation, implying different layers of specificity depending on the cellular context [75]. Moreover, exposition of N2a cells to NO (which is generated in different phases of stroke [76,77]) resulted in increased PrPC production via MEK and p38 signaling, thus implying that stress induction could induce a protective feedback [78].

Shyu et al. reported that ERK1/2 activation precedes increased PrPC expression in a time-dependent fashion in primary cortical cultures subjected to hypoxia-reoxygenation. This was related to neuroprotection, whereas expression of JNK and p38 showed no significant differences. This also was specific as PrPC expression returned to basal levels upon ERK1/2 inhibition. In a rat model of MCAO and bilateral CCA ligation for 90 min, the authors also observed a two-fold increase in PrPC in neurons, astrocytes, and vascular cells of the penumbra. However, the MAPK expression in the brains was not analyzed. Considering their previous work [79,80], they hypothesized that ERK1/2 would activate the heat shock transcription factor-1 (HSTF-1), which would then bind to the heat shock element (HSE) site in the PrPC promoter region, thus upregulating PrPC expression and exerting neuroprotective effects [48]. 

In a study using a different model of stroke (tMCAO for 30 min), PrP0/0 mice had an increased activation of ERK1/2, signal transducer and activator of transcription 1(STAT-1), JNK1/2, and caspase-3 at 3 h after reperfusion, which was neither observed in WT nor in *tga20* mice [49]. Since PrP0/0 mice presented with an increased ischemic volume, the enhanced ERK1/2 phosphorylation was concluded to be detrimental. In addition, 72 h after reperfusion, there was increased expression of JNK1/2 and caspase-3 in PrP0/0 mice, which may relate to delayed cell death [49]. In another study using mice overexpressing human PrPC (TG35), they observed an increase of ERK1/2 in both WT and TG35 mice, albeit lower in TG35 mice, which presented with a neuroprotective phenotype [53]. These authors performed the tMCAO for 90 min. Unfortunately, the lack of immunohistochemical assessment in this study impeded the interpretation of what happens in the infarct core compared to the penumbra, where probably more pro-survival signaling is upregulated.

Because ERK1/2 can participate in both pro-survival and pro-apoptotic signaling, whether its activation has protective (e.g., by upregulating PrPC as an antioxidant) or deleterious effects (i.e., by contributing to oxidative stress damage or by activating apoptotic signals) or if it occurs upstream or downstream of a signaling cascade initiated by PrPC in ischemic damage deserves more studies [81]. Interestingly, neuroprotection conferred by the PrP-N1 fragment was reported to be ERK1/2-signaling independent, but Akt-dependent (see below) [82].

Although PrPC in some pathological conditions can signal through p38 [83,84] that it is upregulated in the penumbra shortly after tMCAO in mice, and together with JNK, plays a role in stroke [73,85], no direct mechanistic link between PrPC and these MAPK has so far been described in the context of cerebral ischemic damage.

### 3.2. The Phosphatidylinositol 3-Kinase/Protein Kinase B/Akt (PI3K/Akt) Pathway and PrPC

As with other signaling pathways, the PI3K/Akt pathway has been implicated in multiple cellular processes ranging from cell survival, angiogenesis, cell growth, and proliferation to apoptosis [86,87,88]. In cerebral ischemia, it is related to neuronal survival, as observed in several models of permanent and tMCAO as well as global models of ischemia in mice, rats, and gerbils (reviewed by [89]). Interestingly, this pathway has been involved in the protection mediated by ischemic pre-/post-conditioning, where smaller ischemic insults, either before or after severe ischemia, induce neuroprotection [90,91].

The first study showing PI3K/Akt pathway activation downstream of PrPC was from Chen et al., who described that this pathway, in cooperation with the MAPK pathway, contributes to PrPC-mediated neurite outgrowth and survival in primary neurons [61]. In another study, PrPC overexpression led to increased PI3K activity, which was neuroprotective, while two neuronal cell lines lacking PrPC and brain lysates of PrP0/0 mice showed decreased PI3K activity. The induction of PI3K by PrPC was copper-dependent, as a copper chelator was decreasing this activation [92]. A protective in vivo role of PrPC via Akt in ischemia was demonstrated by Weise et al., who showed that PrP0/0 mice subjected to transient (60 min) or permanent ischemia presented with a decrease of phosphorylated Akt (pAkt), leading to enhanced caspase-3 activation 6 h after reperfusion when compared to WT mice [51]. However, PrPC-overexpressing TG35 mice (which, as described above, showed reduced ischemic injury that may relate to decreased ERK1/2 activation) did not present increased PI3K/Akt activity after stroke [53]. It is important to note that there is important crosstalk between the ERK1/2 and PI3/Akt pathways during ischemia and reperfusion, with Akt being more activated during the ischemic period and having an inhibitory effect on ERK1/2, whereas ERK1/2 is active during reperfusion without any observed influence back on Akt. Core and penumbra areas also seem to differ in the activation state of these kinases [93]. Therefore, to exactly pinpoint the role of PrPC in the activation of these pathways, more studies analyzing different time points of ischemia/reperfusion, taking different areas into account (i.e., core and penumbra) and ideally differentiating between brain cells, are necessary.

Several studies have identified diabetes as a risk factor of stroke [94] and, together with insulin resistance, these factors lead to increased ischemic damage. In an animal model of diet-induced pre-diabetes (where rats were fed with high-content fructose), the levels of active PI3K/Akt as well as PrPC mRNA and protein levels were downregulated compared with chow-fed animals, pointing to a possible regulation of the *Prnp* gene by the PI3K/Akt pathway [95]. It has also been shown that induction of the insulin-like growth factor 1 ((IGF-1) which has neuroprotective properties in stroke [96]) increased PrPC amounts through PI3K/Akt. Phosphorylation of PI3K/Akt induced phosphorylation of the nuclear transcription factor Forkhead Box O3 (FOXO3a), impairing its binding to the *Prnp* promoter and, thus, negatively regulating PrPC amounts [97]. 

In summary, an increase in PrPC after ischemic damage goes hand in hand with the activation of the PI3K/Akt pathway. Whether it increases PrPC expression first or PrPC induces kinase activation or both (feeding a neuroprotective loop in ischemia), clearly deserves further studies.

### 3.3. The Stress-Inducible Protein-1 (STI-1)

Hop/STI-1 is a co-chaperone that regulates and supports the activity of heat-shock proteins HSP90 and HSP70, which are particularly important for proper protein folding in situations of cellular stress [98]. STI-1 seems to also have additional functions [99], for example, as a ligand of PrPC. This interaction rescues post-mitotic retinal cells exposed to anisomycin or cultured hippocampal neurons exposed to staurosporine from apoptosis [100,101]. This neuroprotective effect is mediated by c-AMP-dependent protein kinase A (cAMP/PKA). Moreover, once STI-1 binds to PrPC, signaling through ERK1/2 leads to neurite outgrowth in cultured hippocampal neurons, an effect that is abolished in hippocampal neurons from PrP0/0 mice [101,102]. STI-1 also promotes SOD activity through PrPC by binding to its N-terminus (i.e., hydrophobic region (HR)), probably by a mechanism involving copper [103]. Moreover, STI-1 can activate the PI3K/Akt/mTOR (Mammalial Target of Rapamycin) pathway through PrPC, triggering an increase in protein synthesis and neuritogenesis and protecting from staurosporine treatment [104]. Thus, it seems possible that most of the known protective functions of PrPC and its related signaling pathways, at least in hippocampal neurons, are mediated by STI-1.

The transmembrane protein that was identified as contributing to this signaling was the α-7-nicotinic acetylcholine receptor (α7nAchR). STI-1-PrPC interaction leading to a Ca2+ influx through α7nAchR is promoted by the physical interaction of the latter with PrPC [105]. Independently of PrPC, STI-1 can signal through other receptors (i.e., Activin receptor-Like Kinase-2 (ALK2) receptors), which can also have an important role in ischemic protection as shown for neurons subjected to OGD [106].

How can the genuinely cytosolic co-chaperone STI-1 interact with cell surface PrPC? Astrocytes can release STI-1 via EVs, where it was shown to be present at the outer leaflet of the membrane and to interact with neurons, triggering PrPC-dependent ERK1/2 signaling [107].

Evidence that the STI-1-PrPC axis plays a role in neuroprotection in ischemic insult came from the experiments by Beraldo et al., showing that transgenic mice expressing only half the amount of STI-1 in astrocytes ((STI-1+/-) note that complete lack of STI-1 is embryonically lethal) had an increased mortality after tMCAO (60 min) compared to WT mice. They also had an increased stroke volume, and surviving animals performed more poorly in behavioral tests than their WT controls. These effects could be attributed to lower EV-dependent STI-1 secretion by the astrocytes and, hence, lack of neuroprotective effects through PrPC [108].

Lee et al. found that STI-1 is increased in the penumbra of patients who died from stroke and in the penumbra, the subventricular region, and the perivascular area of rats subjected to tMCAO [109]. Upregulation was observed in neurons, astrocytes, and endothelial cells. Remarkably, they could show that the hypoxia-inducing factor ((HIFα) a factor involved in the adaptative response to low-oxygen conditions playing an important role in ischemia and also being increased in the penumbra [110,111]) was inducing STI-1 in a time-dependent manner. They also demonstrated in vitro (by transwell migration assays) and in vivo (after tMCAO in WT and PrP0/0 mice, or by lentiviral STI-1 knock-down in mice and rats) that STI-1 led to a PrPC-dependent recruitment of bone marrow-derived cells (BMDC) to the site of injury, which was crucial to the recovery [109]. This role of PrPC in stem cell homing after ischemia will be described in more detail below.

Of note, the neuroprotective effect of STI-1 is not limited to ischemic damage, as it was also observed in Alzheimer disease (AD) where STI-1 can block Amyloid β (Aβ) toxicity by binding to PrPC and eliciting neuroprotection through α7nAchR [112].

### 3.4. PrP Signaling and Glutamate (Receptors)

A consequence of neuronal depolarization induced by the lack of oxygen and glucose is a massive release of glutamate, triggering ionotropic glutamate receptor (iGluR) hyperactivation, mostly the N-methyl-D-aspartate (NMDA) receptors, but also the α-amino-3-hydroxy-5-methyl-4-isoxazole-propionic (AMPA) receptors [113] and, though less studied in stroke, kainate receptors (KAR). Hyperactivation of these receptors results in a cellular influx of Ca2+ that, in turn, activates several signaling pathways causing excitotoxicity and neuronal death [114]. NMDARs are diverse, as they are made up by the combination of four subunits of GluN1, GluN2, and GluN3 (with the most common build-up having two GluN1 with two GluN2 subunits), each one in addition having several variants. GluN1 has eight spliced isoforms encoded by one gene (GluN1a to 4a and GluN1b to 4b). GluN2 has four variants encoded by four genes (A to D) and GluN3 has two members (A and B) encoded by two different genes [115]. Targeting NMDAR as a therapeutic approach for stroke was so far unsuccessful, in part due to NMDARs’ dual role in neuronal death and survival (with deleterious effects at early phases of stroke, while promoting neuronal remodeling subsequently) [116,117,118]. However, targeting the pathways activated by the NMDAR currently represents a promising therapeutic option [119].

PrPC has a role in NMDAR regulation and several lines of investigation point to a role in blocking excitotoxicity [120]. Koshravani et al. demonstrated in hippocampal slices that neurons from PrP0/0 mice had increased synaptic NMDA currents, and PrP0/0 mice with a focal hippocampal injection of NMDA showed enhanced neuronal cell death and increased lesion size compared to WT. They proposed that one function of PrPC is to bind GluN2D (but not to GluN2B) subunits of the NMDAR, thereby silencing it in physiological conditions. In contrast, AMPA receptors were not affected by the lack of PrPC [121].

NMDARs undergo desensitization to avoid Ca2+ overload when overexposed to glutamate or NMDA, but glycine decreases this process in a concentration-dependent fashion [122]. Lack of PrPC or Cu2+ chelation also abrogated desensitization even at physiological glycin concentrations ([123] and reviewed by [124]), indicating that the Cu2+-binding function of PrPC is fundamental for its allosteric regulation of NMDAR. However, further studies by the same group found that, whereas overexpression of mouse PrPC (in *tga20* mice) led to higher NMDAR desensitization (and, thus, more protection by reducing the cation flux), overexpression of human PrPC (in TG650 mice) resulted in the opposite effect (i.e., increasing the sensitivity as seen in PrP0/0 mice). This fact raises doubts about the role of human PrPC in NMDAR regulation but may also simply results from impaired interaction of proteins from different species. AMPA receptor modulation was independent of mouse or human PrPC but dependent on the lack of copper ions [125]. Copper binding to PrPC in the presence of nitric oxide (NO can have both harmful and protective effects after ischemia [126]) induces GluN2A S-nitrosylation, which has a protective effect by decreasing NMDAR activation. The amounts of S-nitrosylation are reduced in PrP0/0 mice, probably contributing to an increased excitotoxic insult observed in these animals [127].

Functional AMPA receptors are composed of the combination of four subunits (GluA1, 2, 3, 4 and GluA5, 6, 7). Its excessive stimulation by glutamate triggers excitotoxicity, and its blockage leads to reduction in ischemic volume [128,129]. In vitro experiments showed that neuronal PrPC downregulated AMPA receptors, contributing to neuroprotection [130]. However, in other instances this modulation is not reproduced [121,125]. Interestingly, binding of PrPC to GluA1 plays a role in the homeostasis of neuronal zinc. Zinc is released together with glutamate at the synapse and acts toxically if not quickly eliminated. PrPC (at least in vitro) behaves as a sensor of zinc, interacting with GluA1 and GluA2 subunits of AMPA receptors to allow for zinc uptake and detoxification of the synapse [131]. As zinc mediates toxicity after stroke through several mechanisms (probably as much as Ca^2+^ does) [132,133], the zinc-PrPC-AMPA receptor axis could be an important mechanism in cerebral ischemia.

The third type of ionotropic glutamate receptors are kainate receptors (KAR) with a relatively poorly defined role in ischemic damage at present [134]. There are five subtypes of KARs (GluK1-5), which function as tetramers and are widely expressed in the nervous system (for an extensive review see [135]). They are expressed pre- and post-synaptically and, among other functions, can modulate glutamate and γ-aminobutyric acid (GABA) release at the pre-synapse. Intravenous and intracerebral injection of kainic acid (KA) leads to severe excitotoxicity and cell death and is used as a model of epilepsy due to its seizure-inducing potential [136,137]. Because PrP0/0 mice present with increased excitotoxicity when challenged with KA, Carulla et al. proposed that PrPC interacts with GluK6/7 receptors at the post-synaptic membrane, modulating JNK3 signaling. When PrPC is absent, GluK6/7 receptors interact instead with postsynaptic density protein 95 (PSD-95) to induce toxic signaling [138]. Although appearing plausible, whether a similar mechanism takes place in ischemic damage remains unknown. As further discussed below, it is important to note that in this study the specific background of PrP0/0 mice used was paramount for the results, which could not be confirmed in other PrP0/0 lines [139,140].

The other group of glutamate receptors is the metabotropic (mGluRs), which play a complex role in ischemic damage, with pro-survival and pro-death effects [134]. They consist of eight subtypes (mGluR1-8), further subdivided into group I (mGluR1 and mGluR5), group II (mGluR2 and mGluR3), and group III (mGluR4 and mGluR6-8), acting as dimers [141]. PrPC can bind to mGluR1 and mGluR5 in vitro after stimulation with a laminin peptide (with laminin being another PrPC binding partner), which causes signaling through phospholipase C (PLC), mobilizing Ca2+ from the endoplasmic reticulum (ER) and triggering neuritogenesis. The binding to one or the other receptor probably depends on availability and localization (e.g., in lipid rafts) [142,143]. The mGluR5-PrPC interaction is also found in a pathological context in vivo, where PrPC binds to Aβ oligomers mediating Fyn signaling through the mGluR5, causing loss of dendritic spines and facilitating long-term depression in AD [144,145]. A similar role for PrPC-mGluR5 interaction in ischemia has not been described.

### 3.5. The Src Family Kinases (SFKs): Critical PrPC Interactors in Neurodegeneration—A Possible Role in Cerebral Ischemia?

When PrPC was proposed as a receptor, the first cell signaling interactor identified was Fyn, a member of the Src family of tyrosine kinases (SFKs) [146] whose phosphorylation was found increased after cross-linking of PrPC with antibodies in the neuronal 1C11 cell line. As Fyn is cytosolic, this interaction was facilitated by PrPC binding to Caveolin-1 [60]. The interaction between PrPC and the neuronal cell adhesion molecule (NCAM) also mediates signals through Fyn to promote neurite outgrowth [147].

It is now well established that one function of PrPC is as a receptor of a wide range of pathogenic β-sheet-rich oligomers associated with neurodegenerative diseases (such as Aβ in AD) [148,149,150]. Through engaging the metabotropic receptor mGluR5 [151], PrPC mediates oligomer-induced toxicity through Fyn kinase. Activated Fyn can phosphorylate the GluN2A and GluN2B subunits of the NMDA receptors, which are then hyperactivated, leading to excitotoxicity with fatal consequences for synapses and neurons [144,152]. Interestingly, the complex of PrPC-mGluR5 is not always engaged in toxic signaling, and the functional outcome rather depends on the ligand. Thus, Aβ oligomers increase toxic signaling, whereas other ligands (such as a peptide mimicking the binding site of laminin) induce endocytosis of the complex, leading to neuronal survival [153]. Another PrPC partner that can mediate Aβ oligomer toxicity via Fyn kinase is the LDL Receptor Related Protein 1 (LRP1) [154].

SFKs, in general, and Fyn, in particular, are also implicated in neuronal ischemic damage [155,156,157,158]. In a model of transient global ischemia in rats, inhibition of SFKs by PP2 led to increased neuronal survival in the penumbra by decreasing tyrosine phosphorylation of GluN2A [159]. Along the same line, inhibition of this Fyn-mediated phosphorylation of GluN2A in a model of neonatal HII (by using transgenic mice with a mutation at the Y1472 phosphorylation site) resulted in decreased neuronal death compared to WT mice [160,161]. In the same model, overexpression of Fyn in turn led to increased brain damage [162]. Inhibition of SFKs by the synthetic inhibitor PP1 resulted in decreased ischemic volume and improved cerebral function in mice 24 h after pMCAO. However, this effect was not seen in Fyn-/- mice [157]. The fact that PrPC acts neuroprotectively in ischemia, and Fyn signaling seems to be related to neuronal death, probably argues against a relevant role of the PrPC–Fyn signaling axis in stroke.

In summary, and as shown in Figure 1, all the signaling pathways’ observations strengthen the idea that PrPC is part of a protein signal multicomplex that, depending on the stimulus and partner (and probably cell type and timing), can be neuroprotective or harmful.

## 4. PrP and Hyperpolarization-Activated Nucleotide-Gated Channel (HCN)

Apart from the above-mentioned signaling pathways, PrPC also participates in the general maintenance of neuronal excitability though HNC, at least in cultured hippocampal neurons, thereby bestowing a general role to PrPC in the regulation of electric signaling in the brain [163]. Astrocytes from mice subjected to pMCAO or rats subjected to global cerebral ischemia occlusion (GIC) increased Hcn1-4 gene expression and protein level in the cortex and hippocampus, respectively, after injury. This altered the electrophysiological properties of the astrocytes after ischemic insult. This mechanism was seen in the recovery phases after stroke (5 weeks) and probably plays an important role in the astrocytic scar formation (one of the typical regenerative processes after stroke [164]) or contributes to reestablishment of ionic homeostasis [165]. However, in mice subjected to tMCAO for 60 min (although showing a decrease of HCN-2 channels in the basal ganglia after 24 h of reperfusion) presence or absence of these channels (as seen after tMCAO in HCN-2-/-mice) seems to have no impact on infarct size and neurological recovery after stroke [166].

## 5. Homing Mechanisms in Cerebral Ischemia: PrP-Related Neurogenesis and Angiogenesis 

After cerebral ischemia, several recovery mechanisms take place, among them the migration of neural progenitor cells (NPCs) from the subventricular zone (SVZ) or the subgranular zone (SGZ) of the dentate gyrus (DG) (and probably from other newly discovery stem cell niches, as well) [167,168]. This phenomenon is not only observed in several models of ischemic stroke but also in human brains [169,170]. Moreover, there is vessel remodeling and new vessel formation in the penumbra [171], which is also confirmed in humans [172]. Bone marrow-derived cells (BMDCs) are another potential source of new endothelial cells and neurons after recruitment to (“homing”) and differentiation in the damaged brain tissue [173,174,175]. PrPC has been implicated in embryonic and adult neurogenesis. In vitro experiments demonstrated a role in differentiation as PrP0/0 NPCs remain much longer undifferentiated than those isolated from WT or PrPC-overexpressing mice. In vivo, mice overexpressing PrPC had more cellular proliferation in the SVZ and DG than WT or PrP0/0 mice. However, in steady-state conditions, increased NPCs’ proliferation did not lead to a difference in the net amount of new cells, indicating that other factors are related to NPC survival [176]. The role of PrPC in neuronal proliferation and survival has also been demonstrated after olfactory sensory epithelium injury in vivo, highlighting the importance of PrPC in the process of neuronal regeneration [177].

A similar role for PrPC is proposed after stroke. By studying PrPC-overexpressing mice after tMCAO (45 min) and long-term reperfusion (28 days), Doeppner et al. observed improved neurological recovery compared to WT and PrP0/0 mice that matched with increased neurogenesis and angiogenesis [54]. They demonstrated that PrPC overexpression led to decreased proteasome activity (24 h after reperfusion) and to decreased ERK1/2 activation [53,178,179]. As a consequence of reduced proteasomal activity, the Hypoxia-Inducible Factor-1α (HIF-α, which promotes differentiation of neural stem and progenitor cells (NSPCs) [180]) and the pituitary adenylate cyclase-activating polypeptide ((PACAP38) also related to neuroprotection in stroke [181]) were not degraded. Homing of systemically injected NPCs to the peri-infarct site was enhanced, probably through increased HIF-α and PACAP38 levels [54,182]. Intriguingly, PrP0/0 mice also showed increased neurogenesis compared to WT. However, this was likely a consequence of the bigger injury and did not improve overall recovery [54].

Further investigations by Lin et al. showed that PACAP38 was increased in neurons, glia, and endothelial cells in the penumbra of rats subjected to transient ischemia (90 min) and in penumbral neurons in human brains after stroke. This upregulation was a consequence of HIF-α binding to the PACAP38 promoter. In vitro, CD34+BDMCs (presenting high amounts of PAC1, the PACAP38 receptor) migrated in a PACAP38 concentration-dependent fashion, which was further depending on CD34+BDMCs upregulation of PrPC, α-6 integrin, and the matrix metalloproteinases 2 and 9 (MMP2, and MMP9), among other proteins. In vivo, PACAP38-treated PAC1+/+/EGFP chimeric mice showed stronger BMDC homing than PACAP38-treated PAC1−/−EGFP or saline-treated control mice, together with significantly reduced infarct volumes [182]. Moreover, brain blood vessels secreted PACAP38 and showed upregulated PrPC and α-6 integrin. This probably stimulates the cell adhesion to the laminin present at the surface of the vessels (or maybe PrPC-PrPC homophilic interactions) facilitating migration. When exogenous PACAP38 was administered to stroked rats, smaller infarcts, better neurological outcome, enhanced glucose metabolism, decreased cell death, and increased blood vessel density and cerebral blood flow was observed in the ischemic penumbra [182]. Thus, PrPC upregulation in endothelial cells in the penumbra and in CD34+BMDCS seems to influence angiogenesis through a pathway involving PACAP38-PAC1.

Another way by which PrPC might be involved in neuronal stem cell migration, is through STI-1 signaling. As pointed out above, Lee et al. described that STI-1 is increased in neurons, glia, and endothelial cells after stroke in rats and humans. This increase was mediated by HIF-α binding to the STI-1 promoter. Notably, PrP0/0 mice had fewer BMDCs recruited to the ischemic area and showed increased injury volume than WT mice tMCAO (120 min) three days after reperfusion following tMCAO (120 min). Lentiviral silencing or increasing STI-1 expression caused BMDC recruitment prevention or promotion, respectively. Recruitment was dependent on the MMP2/9 activity, as their experimental inhibition also decreased the recruitment and neurological recovery [109].

Of note, binding of STI-1 to PrPC also plays a role in proliferation and self-renewal of NPCs [183] and arachnoid-pia stem cells (APSCs) present in the leptomeninges; both could potentially contribute to neurogenesis after stroke [184,185,186,187]. 

## 6. Proteolytic Fragmentation Enables Functional Diversity–PrPC Derivatives and Stroke

One obvious question arises when considering the variety of potential implications of PrPC discussed above merely in the context of ischemia: How can one protein alone be linked with so many (suggested) functions? It is tempting to speculate and increasingly appreciated that evolutionarily conserved proteolytic cleavages, some of them known for decades [188,189,190,191], may partially account for this [29,192].

PrPC can undergo different site-specific cleavages: (1) α-cleavage in the middle of its sequence generates soluble N1 and membrane-bound C1 fragments [189,190,193]. Though being regarded as the major physiological cleavage event, the responsible protease(s) is (are) not identified [194,195]. (2) β-cleavage taking place around residue 90 and separating soluble N2 from membrane-attached C2 fragments is rather linked with pathological conditions such as oxidative stress and may represent a protective feedback mechanism [196,197,198]. (3) The γ-cleavage, with a presumed cleavage site between residues 176 and 200, releases a large, non-glycosylated N3 fragment, while a short C3 fragment remains at the membrane [199,200]. Considering that γ-cleavage of PrPC was only described a few years ago, it would not be too surprising if additional, potentially tissue-specific cleavage events exist that went undetected so far. (4) A proteolytic cleavage even closer to the GPI anchor mediated by A Disintegrin And Metalloprotease domain-containing protein 10 (ADAM10) results in shedding of preferentially diglycosylated PrP from the cell surface [201,202,203]. Reliable characterization of PrPC and its various fragments (and screening for alterations, e.g., upon pharmacological intervention) may profit from a recently described method based on capillary Western blotting [204].

While interest in intrinsic biological functions of PrPC cleavage fragments in different tissues is growing [29,205], systematic studies are scarce. Some of the already described effects, however, may well relate to protective and regenerative processes following ischemic insults. 

The biological functions of soluble PrPC in body fluids or the brain parenchyma are not clear to date. Some in vitro studies have used recombinant PrP (recPrP), which, despite lacking the glycans, may be considered a suitable analog of shed PrP (sPrP). A recent study revealed that recPrP acts as a ligand on mesenchymal stem cells, inducing ERK1/2 and Akt signaling and supporting neuronal differentiation [206]. Amin et al. have shown that recPrP supports neurite outgrowth as a chemotactic stimulus guiding the axonal growth cone [207]. Notably, both effects seem to depend on the presence of PrPC at the recipient cell’s surface, indicating the requirement of homophilic ligand–receptor interaction. Moreover, sPrP/recPrP has recently been implicated in an immunological chemokine cascade, resulting in increased recruitment of monocytes into brains of patients suffering from HIV neuropathogenesis [208]. Though seemingly unrelated at first glance, these studies already indicate the involvement of sPrP in regulating intercellular communication in various processes with likely relevance also to stroke.

More data are available for the fragments resulting from the α- and β-cleavage of PrPC. Several studies by Haigh, Collins, and colleagues have shown protective effects in cellular stress conditions conferred by the N1 and N2 fragments released upon α- and β-cleavage, respectively [209,210,211]. Also, the quiescence of neural stem cells was reported to be modulated by these fragments [212] with potential relevance to adult neurogenesis upon stroke. The corresponding membrane-bound *C*-terminal fragments have been implicated in regulating p53-dependant apoptosis and cell survival [213]. As discussed in the following section, recent findings from our group suggest that the C1 fragment could affect the interaction between EVs and their target cells and, thus, intercellular information exchange.

A major clue pointing towards the possibility that PrP-associated protective effects in ischemic conditions (discussed in previous paragraphs) might not directly be linked with membrane-bound full-length PrPC but rather with its proteolytically released fragments comes from a study of Checler and colleagues [82]. In this study, N1 was shown to protect neurons and rat retinas from ischemic damage. More recently, the same fragment was shown to regulate the interaction of microglia with other brain cell types [214].

Lastly, while PrPC cleavages regulate PrP-associated ERK1/2 signaling [215], likely influencing functional interaction of PrPC with diverse binding partners and generating fragments acting as biologically active molecules in trans or in distance, the *C*-terminal shedding may be a mechanism to terminate functional interactions in cis [29].

At this point, it remains to be further investigated if, indeed, particular PrPC fragments, rather than cell surface PrPC, are responsible for the suggested PrP-associated effects on oxidative stress and neuroprotection, neurito- and angiogenesis, neural regeneration, and adult neurogenesis. However, increasing evidence linking PrPC fragments with intercellular communication strongly suggests this view. Although further studies on the role of PrPC cleavages and fragments in stroke are clearly required, identification and pharmacological stimulation of endogenous proteases, as well as targeted design and therapeutic administration of protective PrPC derivatives, may represent new treatment options in ischemic conditions.

## 7. Another Way of Releasing Information: EVs, PrP and Its C1 Fragment, and Stroke

In 1997 a study by Schätzl et al. showed that the conditioned medium of prion-infected cells in culture was transferring infectivity to naive cells by a back-then not identified mechanism [216]. It was in 2004 that this transfer process was clarified when both PrPC and its misfolded isoform, PrPSc, were found in exosomes isolated from the media supernatant of a prion-infected neuronal cell line [217,218,219,220,221].

Exosomes are a subtype of EVs, a term that includes a heterogeneous group of double-membrane vesicles released into the extracellular space by possibly all types of cells. It is known that EVs have a central role in cell-to-cell communication because they transport and deliver cargos, such as lipids, proteins, mRNA, and noncoding RNAs, from one cell to another and, thereby, to regulate gene expression [222]. EVs are categorized according to their origin/biogenesis and size, although it is clear that strict boundaries among the EVs are not easy to draw [223,224,225,226,227]. Classically, EVs are differentiated in apoptotic bodies (diameter between 500 nm and 5 µm) originating from cells undergoing apoptosis, microvesicles (100–1000 nm), budding from the plasma membrane, and the above-mentioned exosomes (30–150 nm) formed in multivesicular bodies (MVB) [228,229]. 

PrPC is enriched in EVs [230,231,232] but not much is known about its physiological functions therein. We recently found that EVs purified from mouse brain are highly enriched in PrPC and its proteolytic C1 fragment (see paragraph above) [232]. PrP levels were further increased in EVs isolated from mice subjected to tMCAO (40 min) at 24 h after reperfusion. We showed that presence or absence of PrPC on these vesicles was influencing the delivery mechanism to recipient cells, and we proposed that PrP may determine the destination of cargo by an uptake-versus-fusion decision.

We also demonstrated that, after tMCAO, the major brain cell population contributing to the total EVs’ pool shifts from microglia to astrocytes. Interestingly, Guitart et al. demonstrated in vitro that astrocytes release exosomes packed with PrPC and that, under ischemic conditions (OGD), increased release had protective effects in primary neurons [233]. If this protective effect is due to enhanced tethering to target cells and delivery of protective molecules deserves further investigations.

The role of EVs from different cell types and their contribution to the ischemic damage and recovery is still not clear, even though different studies have tried to clarify this certainly complex scenario (reviewed in [6]).

EVs represent an attractive therapeutic tool for neuronal diseases, mainly because of their ability to cross the blood–brain barrier (BBB), their low immunogenicity, and the fact that they can be easily engineered to target certain cells and/or loaded with therapeutically active compounds [234,235]. EVs can be modified to specifically target brain cells after being injected intravenously [236]. Since EVs are present in many body fluids, they are also very attractive as potential biomarkers.

For stroke, a currently major focus is on therapy with EVs purified from mesenchymal stem cells (MSCs), as they improve neurogenesis, angiogenesis, and long-term recovery in mice when administrated intravenously after tMCAO [237,238,239,240,241,242]. However, many unknowns regarding the EVs’ isolation or best cell source together with their biodistribution (which is influenced depending on the administration route [243]) and biostability/half-life [244,245] have to still be determined.

Recently, as a proof of principle that EVs’ regenerative potential is tissue-independent, it has been shown that EVs derived from NPCs improve neuronal recovery after tMCAO in mice similarly to MSCs [246].

## 8. Conclusions

PrPC is suggested to be involved in an overwhelming number of (patho)physiological mechanisms, and it is important to be careful in the interpretation of individual findings. Two different PrP0/0 mouse lines have classically been used in many of the experiments: The Zurich mice ((ZrchI) the first one generated [33]) and the Edinburgh mice (Edbg or Npu), with both of them not presenting gross abnormalities and rather normal behavior. Two other lines, the Nagasaki (Ngsk) and Rcm0 lines, developed ataxia and neurodegeneration in aged mice, a phenomenon that was later attributed to the abnormal expression of the Doppel protein (Dpl) as a consequence of the genetic knock-out strategy [247,248,249]. This was further reproduced and demonstrated by the generation of the ZurichII (ZrchII) mice [250]. A conditional mouse model using the *Cre-loxP* system and allowing for the depletion of PrPC post-natally showed alterations in neuronal excitability but no overt neurodegeneration [251]. The fact that the PrP0/0 mice (with the exception of the Edbg line) were generated with a mixed background (C57BL/6x129sv(ev)) poses the question whether many of the phenotypes observed are a consequence of polymorphisms and variations of the *Prnp* flanking genes, rather than of the absence of PrPC itself [27]. For example, as mentioned above, Carulla et al. found that, although PrPC was neuroprotective against kainate excitotoxicity, some unidentified flanking genes dependent on the mixed genetic background were also playing a role and some mouse lines were, thus, not suitable for the experiments [140]. A novel mouse line with a pure C57BL/6 background has more recently been generated (ZrchIII), which offers the opportunity to better control inconsistencies due to genetic artifacts and, thus, better dissect ‘true’ functions of PrPC [252]. Consequently, it would be advisable that key experiments addressing the neuroprotective role of PrPC in stroke are validated using this new model.

Moreover, in stroke research, the different procedures of ischemic damage induction and durations (e.g., of vessel occlusion, reperfusion time, or time points of assessment) in different species or lines make a comparison between studies extremely difficult. This is further complicated by a non-standardized read-out regarding behavioral outcome, which may lead to rather ‘subjective’ results [253].

Nevertheless, the fact that different species and different paradigms of ischemic damage induction consistently indicate PrPC involvement in neuroprotection advocates for important roles of PrP in stroke, probably conducted by different mechanisms, as shown in Figure 2. Much more work is needed to clearly pinpoint the exact role(s) of PrPC in stroke and to potentially devise novel PrP-directed or even PrP-based therapeutic options. However, it is conceivable that the manipulation of PrPC amounts (or its C1 fragment) in EVs improves the delivery of therapeutic agents to recipient cells (e.g., neurons at the penumbra); and as sPrP and N1 have been involved in neurite outgrowth and neuroprotection (as mentioned above), stimulation of the production of these fragments could also have therapeutic potential in ischemic stroke.

All in all, different lines of research indicate PrPC as a pivotal player in ischemic damage and highlight its therapeutic potential.

## Figures and Tables

**Figure 1 cells-09-01609-f001:**
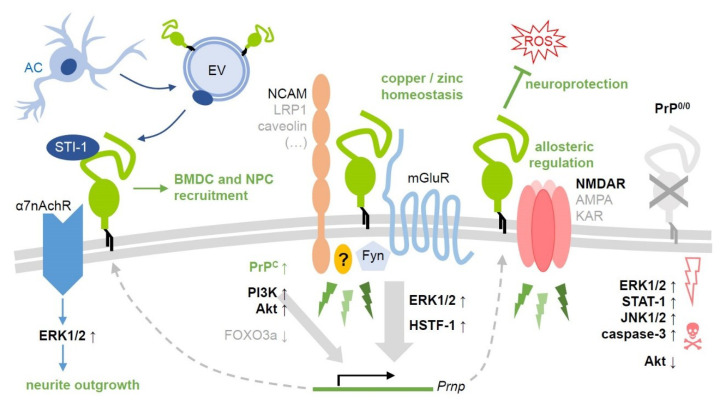
Selected PrPC-related signaling pathways and their (potential) relevance after ischemic insults. As a GPI-anchored protein, PrPC (center, green) is unable to transmit signals into the cell. To do so, it forms cell type- and context-dependent complexes with various transmembrane (NCAM, LRP1, caveolin, subunits of mGluR, NMDA, AMPA, kainate, or a7nAch receptors) and cytosolic signaling proteins. This allows for its involvement in a variety of signaling events (green thunderbolts) with diverse functional outcome. Some of those have been investigated in detail, while others remain rather poorly defined. Here, only a few cascades with (likely) implications in stroke are depicted (with no claim to be complete). These are mostly associated with neuroprotective and regenerative processes in the penumbra following ischemic damage. Note that small upward arrows next to the protein names indicate upregulation/activation, whereas downward arrows stand for downregulation/inhibition (ROS: Harmful reactive oxygen species). Some factors, such as the MAP kinase ERK1/2, may play a dual role causing both beneficial and detrimental effects (e.g., in PrP0/0 mice, on the very right), depending on the exact region, cell type, and time point after damage induction as well as on the experimental paradigm. Activation of the protein Akt has so far mostly been associated with protective effects in this context. Intercellular communication is of utmost importance in regenerative processes after stroke, as exemplified here by the co-chaperone STI-1, which is released (possibly in association with extracellular vesicles, EV) by astrocytes and binds to cell surface PrPC. Consequences of this interaction may include recruitment of bone marrow-derived (BMDC) and neural precursor cells (NPC) to the damaged brain region as well as neurite outgrowth. References given in the main text.

**Figure 2 cells-09-01609-f002:**
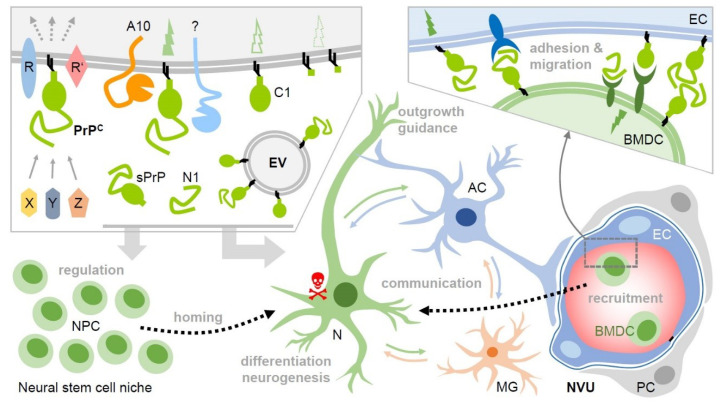
Potential roles of membrane-bound and released PrP (fragments) in intercellular communication and regenerative processes upon stroke. Upper left box: Various soluble ligands (X, Y, Z) and interaction partners have been described to bind to cell surface PrPC, which then, in complex with other membrane proteins (R, R’), induces cell type- and context-dependent signaling with diverse functional consequences. PrPC undergoes conserved cleavages at specific sites by ADAM10 (A10) and other proteases. This creates truncated membrane-bound forms (e.g., C1) and may modify or terminate PrP’s functions on the very same cell. Of note, soluble fragments, such as shed PrP (sPrP) or N1 are generated. Together with PrP enriched on and released via extracellular vesicles (EVs), this creates a pool of diverse PrP versions in the extracellular space. Upon stroke, PrP expression is upregulated in the penumbra and cell surface PrP (fragments) as part of signaling-competent receptor complexes and released forms (acting as bioactive ligands) may play decisive roles in neuroprotection and regeneration. For instance, they play a role in the regulation of stem cells, particular neural precursor cells (NPCs), and may be involved in the recruitment (‘homing’) to the site of hypoxic injury and differentiation/neurogenesis therein. PrP-associated neuroprotection may also relate to stimulated communication between neurons (N), astrocytes (AC), microglia (MG), and other cell types. This may, among other aspects, induce cellular repair mechanisms in support of damaged neurons (red skull) and support axon guidance and neurite outgrowth. PrP and its fragments also seem to be critically involved in the recruitment of bone marrow-derived cells (BMDC) from the periphery, which, similarly to NPCs, may replenish cells lost in the damaged brain tissue. They may also support angiogenesis (not depicted here). Again, this could be associated with PrP versions acting as ligands and/or receptors (even homophilic interactions are conceivable), allowing BMDC to dock to endothelial cells (EC) and migrate through the vessel into the damaged tissue (see bottom left part and box above; NVU: neurovascular unit; PC: pericyte). Note that many of these aspects are speculative at present and that several other molecules clearly play important roles in these processes. This scheme aims to present a PrP-focused view to simplify matters. Key references are provided in the main text.

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
