# Peer review of "Show Me Your Friends and I Tell You Who You Are: The Many Facets of Prion Protein in Stroke"

_cells, 2020, doi:10.3390/cells9071609_

Round 1

Reviewer 1 Report

The manuscript “Show Me Your Friends and I Tell You Who You Are: The Many Facets of Prion Protein in Stroke” by Puig and colleagues review the advances in mechanisms underlying PrPC-induced neuroprotection. This is a high-quality review with constructive analysis of the literatures. I only have minor concerns.

The authors could include a more comprehensive discussion of the therapeutic potentials such as therapeutic time window and potential side effects, etc.

Author Response

The manuscript “Show Me Your Friends and I Tell You Who You Are: The Many Facets of Prion Protein in Stroke” by Puig and colleagues review the advances in mechanisms underlying PrPC-induced neuroprotection. This is a high-quality review with constructive analysis of the literatures. I only have minor concerns.

The authors could include a more comprehensive discussion of the therapeutic potentials such as therapeutic time window and potential side effects, etc.

We are very grateful to the reviewer for the nice comments. Because of its clear involvement in neuroprotection, we believe that PrPC could be envisaged as a potential therapy in ischemic stroke in the future. However, as the mechanisms involved in its upregulation in stroke and the signaling cascades triggered by PrPC are still poorly defined, it is currently difficult to draw a picture of its therapeutic potentials. We have, though, extended a bit on what was already suggested in the conclusions (lines 622-629, in red).

Reviewer 2 Report

This paper represents an extensively elaborated review on very important subject with the valuable significance to the human neurology. The structure of the paper is well prepared, however the scientific merit and informational potency could be improved if authors could elaborate to more details different models of brain ischemia. Paper describes mostly MCAO model of ischemic damage and its impact to the development of the disorder. In fact, the hypoperfusion models might have a high relevance regarding to the clinic, and also global ischemic models and/or with the combination of metabolic toxicants(risk factors) such as homocysteine potently proved causal linkage between insult and progression of the disease.

Reviewer also suggest to cite more original papers instead of review papers.

Author Response

This paper represents an extensively elaborated review on very important subject with a valuable significance to the human neurology. The structure of the paper is well prepared, however the scientific merit and informational potency could be improved if authors could elaborate to more details different models of brain ischemia. Paper describes mostly MCAO model of ischemic damage and its impact to the development of the disorder. In fact, the hypoperfusion models might have a high relevance regarding to the clinic, and also global ischemic models and/or with the combination of metabolic toxicants(risk factors) such as homocysteine potently proved causal linkage between insult and progression of the disease.

We thank this reviewer very much for the valuable insight. We agree that hypoperfusion is a clinically relevant issue that can occur in situations with severe stenosis of coronal arteries. However, the review is focused on the role of PrPC in the many aspects of the pathophysiology of ischemic stroke, and, unfortunately, we could not find any study (neither regarding hypoperfusion models and PrPC nor the effects of homocysteine linked to PrPC) in stroke. In view of the already “comprehensive” nature of this review, we would rather refrain from including additional aspects that have not been associated with the overall topic of PrPC.

Reviewer also suggest to cite more original papers instead of review papers.

We agree that there are several reviews referenced here, but we believe that they give to the reader a good overview of specific topics that we are not developing further. Importantly, the relevant papers directly dealing with PrP are original.

Reviewer 3 Report

Berta Puig & colleagues have provided a review of the cellular prion protein (PrPC) with respect to stroke. PrPC is an enigmatic protein, and its functions in health & disease have remained a ‘hot topic’ in neurology and biochemistry for several years (this interest is not waning anytime soon). This Review is timely, as the occurrence of stroke allows insights into the enigmatic roles of PrPC. Separately, the Review is valuable as a contribution to appreciating the etiology of stroke that can inspire readers regarding treatments.

The Review is accompanied by beautiful schematics that are very informative and clear. The writing is sounds and easy to follow. At certain spots, noted below, the thesis of the writing could be made more explicit.

Minor points:

  1. Line 64-66: the argument from conservation in support of an important function is very meritorious. Some nuances: prion protein is extraordinarily well-conserved within the mammals (not all of evolution). Within the vertebrates ALL the protein domains are conserved (e.g. Cotto…Babin 2005). The references here could send the reader to papers from the Allison lab, that shows the conservation of function not just by sequence or domain similarity, but where loss of PrPC function in zebrafish shows conserved phenotypes to KO mice. Also - Some of these conserved physiological roles are probably germane to stroke- control of excitotoxicity, NMDARs, mGluR5, signalling to Fyn kinases (see Malaga-Trillo). I am incredulous that reference 23 is the best one to use here.
  2. Section 3 might incorporate the manuscript by Daude et al 2105 in Prion PMID: 26516793. It uses PRNP KO mice (and shadoo KO mice) in an ischemic model and speculates on roles for the N-terminal region.
  3. Line 138 – There is room to be more critical of the literature, or at least somewhat less conclusive. A skeptical view would be that the PRNP KO mice have a linked alteration. The PRNP KO mice of various derivations show quite different phenotypes, in part because some have substantial confounds (e.g. Carulla et al 2015 Scientific Reports comes to mind). Perhaps worth a note here, then, if the stroke susceptibility has been tested across multiple versions of PRNP KO or on multiple backgrounds? [ I now see you have built these concepts into the section 8 Conclusions – kudos!]

Trivial stuff:

  1. Line 11 – I would’ve guess the relevance of PrPC to stroke was mostly about the later phases of stroke management, rather than the acute phase. But I am no clinician…. Ideas on line 51 support that PrPC is more relevant after the acute events? Oh, as I read on line 126 speculates it might be both (maybe that is a better message for the Abstract?)
  2. line 121 – “found” s/b “find”
  • line 143 – this Section & subsection would both benefit from a topic sentence. What is the goal of this section 3? A comprehensive list of past papers considering the topic? A critical review with suggestions of key missing experiments to be prioritized by the field?
  1. Line 373 – are some of these ligands altered in stroke? E.g. amyloid beta oligomers?
  2. Section 6 the authors might enjoy a recent paper by Castle et al 2019 JBC PMID: 30578300.

Author Response

Berta Puig & colleagues have provided a review of the cellular prion protein (PrPC) with respect to stroke. PrPC is an enigmatic protein, and its functions in health & disease have remained a ‘hot topic’ in neurology and biochemistry for several years (this interest is not waning anytime soon). This Review is timely, as the occurrence of stroke allows insights into the enigmatic roles of PrPC. Separately, the Review is valuable as a contribution to appreciating the etiology of stroke that can inspire readers regarding treatments.

The Review is accompanied by beautiful schematics that are very informative and clear. The writing is sounds and easy to follow.

We are very thankful to the reviewer for the appreciation of our review.

 At certain spots, noted below, the thesis of the writing could be made more explicit.

Minor points:

  1. Line 64-66: the argument from conservation in support of an important function is very meritorious. Some nuances: prion protein is extraordinarily well-conserved within the mammals (not all of evolution). Within the vertebrates ALL the protein domains are conserved (e.g. Cotto…Babin 2005). The references here could send the reader to papers from the Allison lab, that shows the conservation of function not just by sequence or domain similarity, but where loss of PrPC function in zebrafish shows conserved phenotypes to KO mice. Also - Some of these conserved physiological roles are probably germane to stroke- control of excitotoxicity, NMDARs, mGluR5, signalling to Fyn kinases (see Malaga-Trillo). I am incredulous that reference 23 is the best one to use here.

We are sorry to realize that reference 23 was indeed wrong and unintended, and we are thankful that the reviewer has seen this mistake. As kindly suggested, we have added the references of Cotto et al., Allison et al., and Malaga-Trillo and colleagues. We also added the reference of Wopfner et al. referring to the high degree of conservation between mammals (but not between mammals and birds). Changes are in red (lines 63-65).

  1. Section 3 might incorporate the manuscript by Daude et al 2105 in Prion PMID: 26516793. It uses PRNP KO mice (and shadoo KO mice) in an ischemic model and speculates on roles for the N-terminal region.

We thank the reviewer for this suggestion and we have now referenced and added a comment on this paper (lines 116 to 118, in red).

  1. Line 138 – There is room to be more critical of the literature, or at least somewhat less conclusive. A skeptical view would be that the PRNP KO mice have a linked alteration. The PRNP KO mice of various derivations show quite different phenotypes, in part because some have substantial confounds (e.g. Carulla et al 2015 Scientific Reports comes to mind). Perhaps worth a note here, then, if the stroke susceptibility has been tested across multiple versions of PRNP KO or on multiple backgrounds? [ I now see you have built these concepts into the section 8 Conclusions – kudos!]

We agree that this is an important aspect, which (as also noted by this reviewer) we decided to discuss in the final “Conclusions” section.

 Trivial stuff:

  1. Line 11 – I would’ve guess the relevance of PrPC to stroke was mostly about the later phases of stroke management, rather than the acute phase. But I am no clinician…. Ideas on line 51 support that PrPC is more relevant after the acute events? Oh, as I read on line 126 speculates it might be both (maybe that is a better message for the Abstract?)

In the abstract, we start with „there is no treatment for the acute phase of stroke“, and we mention afterwards that „There is a need to search for therapeutic options to promote neurological recovery after stroke“. We think that with these two sentences we cover both time-points where PrPC would potentially be relevant.

line 121 – “found” s/b “find”

We apologize for this mistake, which is now amended (in red, now line 124).

line 143 – this Section & subsection would both benefit from a topic sentence. What is the goal of this section 3? A comprehensive list of past papers considering the topic? A critical review with suggestions of key missing experiments to be prioritized by the field?

We are thankful for this comment and have modified the title of this section 3 (now in red). The aim of this comprehensive overview indeed is to present known PrP-related signaling pathways and discuss them with regard to their confirmed or potential involvement in ischemia. While some aspects are experimentally worked out in more detail, other aspects are –at present- rather poorly understood and thus have a rather speculative character (which we hope to have emphasized as such). We have sorted the subsections by individual cascades for a better understanding of prospective readers (although -of course- cascades are partially connected/interfering and in the pathophysiological setting not mutually exclusive). Though trying to put some order and categorization, the complexity in reality is also indicated to some degree in accompanying Figure 1.

Line 373 – are some of these ligands altered in stroke? E.g. amyloid beta oligomers?

Increased A-β is found in the plasma of patients suffering from acute stroke compared to controls (Lee et al. DOI: 10.1007/s00702-004-0274-0). Interestingly, A-β (mainly in its non-aggregated form, but also as extracellular plaques) accumulates in blood vessels in the nearby ischemic areas after tMCAO, thereby causing damage (Henrique Martins et al. DOI: 10.3390/biom9080350).  However, while these aspects are interesting, we would rather refrain from further extending(or changing) the focus of this already pretty comprehensive review and hope for this reviewer`s understanding.

Section 6 the authors might enjoy a recent paper by Castle et al 2019 JBC PMID: 30578300.

Thank you very much for the suggestion. This paper indeed presents a very interesting method that may become a new standard when assessing proteolytic fragmentation (and its manipulation or consequences). We have now added this reference in the respective section (lines 506-508). By the way: Related to the higher PrP molecular weight revealed with the capillary western blot, we see something similar when using the precasted gels of Invitrogen (NuPage) with commercial buffers.